# Multivariate Independent Component Analysis Identifies Patients in Newborn Screening Equally to Adjusted Reference Ranges

**DOI:** 10.3390/ijns9040060

**Published:** 2023-10-20

**Authors:** Štěpán Kouřil, Julie de Sousa, Kamila Fačevicová, Alžběta Gardlo, Christoph Muehlmann, Klaus Nordhausen, David Friedecký, Tomáš Adam

**Affiliations:** 1Department of Clinical Biochemistry, University Hospital Olomouc, 779 00 Olomouc, Czech Republicdavid.friedecky@fnol.cz (D.F.); 2Laboratory of Metabolomics, Institute of Molecular and Translational Medicine, Palacký University Olomouc, 779 00 Olomouc, Czech Republic; 3Department of Mathematical Analysis and Applications of Mathematics, Faculty of Science, Palacký University Olomouc, 779 00 Olomouc, Czech Republic; kamila.facevicova@upol.cz; 4Institute of Statistics & Mathematical Methods in Economics, Vienna University of Technology, 1040 Vienna, Austria; 5Department of Mathematics and Statistics, University of Jyväskylä, 40014 Jyväskylä, Finland; 6Faculty of Health Care, The Slovak Medical University in Bratislava, 974 05 Banská Bystrica, Slovakia

**Keywords:** newborn screening, independent component analysis, mass spectrometry, multivariate statistical analysis, inborn errors of metabolism, compositional data analysis

## Abstract

Newborn screening (NBS) of inborn errors of metabolism (IEMs) is based on the reference ranges established on a healthy newborn population using quantile statistics of molar concentrations of biomarkers and their ratios. The aim of this paper is to investigate whether multivariate independent component analysis (ICA) is a useful tool for the analysis of NBS data, and also to address the structure of the calculated ICA scores. NBS data were obtained from a routine NBS program performed between 2013 and 2022. ICA was tested on 10,213/150 free-diseased controls and 77/20 patients (9/3 different IEMs) in the discovery/validation phases, respectively. The same model computed during the discovery phase was used in the validation phase to confirm its validity. The plots of ICA scores were constructed, and the results were evaluated based on 5sd levels. Patient samples from 7/3 different diseases were clearly identified as 5sd-outlying from control groups in both phases of the study. Two IEMs containing only one patient each were separated at the 3sd level in the discovery phase. Moreover, in one latent variable, the effect of neonatal birth weight was evident. The results strongly suggest that ICA, together with an interpretation derived from values of the “average member of the score structure”, is generally applicable and has the potential to be included in the decision process in the NBS program.

## 1. Introduction

Inborn errors of metabolism (IEMs) are caused by an enzyme deficiency that usually leads to the accumulation of substrates of the defective enzyme. Nearly 1900 diseases are currently classified in this group. Newborn screening (NBS) is an active and widespread search for diseases in their early, preclinical stages so that these diseases can be diagnosed and treated before they become manifest and cause irreversible damage to a child’s health.

In the screening of IEMs, flow injection–tandem mass spectrometry analysis is used to quantify specific analytes (amino acids, acylcarnitines) which are diagnostically relevant to the diseases in question [1]. The data evaluation is performed using reference ranges established on a healthy newborn population using quantile statistics of untransformed data. The key diagnostic biomarkers of IEMs are elevated levels of substrates of defective enzymes. The first introduced screening program detected phenylketonuria (PKU) patients using a bacterial inhibition assay for (semi)quantitative analysis of phenylalanine (Phe) in dried blood spots [2]. Besides reference ranges for individual diagnostic metabolites, their mutual ratios [3] are applied as a post-analytical tool. The cutoff target ranges of analytes and ratios are then defined as selected percentiles of the control and diseased populations. When overlaps occur, adjustments are made to optimize sensitivity and specificity considering Wilson and Jungner criteria [4].

The first application of amino acid relations used in diagnostics was the phenylalanine/tyrosine (Phe/Tyr) ratio (substrate and product of the defective enzymatic conversion, respectively) for the detection of PKU heterozygous carriers [5]. This approach was successfully applied to NBS, lowering substantially (to 25%) false positives [6,7] compared to evaluation based on the absolute concentration of Phe alone. Since its introduction, several ratios were suggested for screening of medium-chain acyl-CoA dehydrogenase deficiency (MCAD) [8] and/or carnitine palmitoyltransferase II deficiency (CPTII) [9,10]. Similarly, ratios for screening pyruvate dehydrogenase complex deficiencies and other mitochondrial disorders associated with lactic acidosis and variably elevated alanine (Ala) and proline (Pro) levels have recently been proposed [11]. The authors used strictly ketogenic amino acid leucine (Leu) and lysine (Lys), not involved in the glycolytic pathway generating pyruvate, for “normalizing metabolites in quantitative analysis of Ala and Pro” [11] and forming diagnostically effective Ala/Leu and Pro/Leu ratios. In the field of IEMs, the developed diagnostic ratios generally correct for sampling variability (including the hematocrit effect) and elucidate substrate/product relations (e.g., proximal Phe/Tyr ratio; distant in case of linear enzymatic pathways, e.g., octanoylcarnitine (C8) and acetylcarnitine (C2) acylcarnitines). In the Czech Republic, the NBS program targets 15 IEMs listed in Table 1.

In general, it is expected that the metabolic findings in patients are atypical compared to the healthy population. For this reason, the data from patients can be understood as outlying observations. The above-described routine approach to NBS, based on reference ranges of individual metabolites or their mutual ratios, uses only marginal information from the metabolic profile, which technically represents a multivariate observation. Therefore, it essentially provides an incomplete picture, and univariate outlier detection methods are missing (for example, outliers where the interplay between variables is atypical). Classical multivariate outlier detection tools are usually based on the Mahalanobis distance which, however, scales badly with increasing dimension [12]. Hence, given the dimensionality of the NBS data, dimension reduction prior to outlier detection seems a natural way to proceed. On the other hand, traditionally used unsupervised dimension reduction methods like principal component analysis (PCA) might not always be suitable as, for example, outliers do not necessarily need to be found in the direction of high variations, representing the main aim of the method.

The aim of this study is the application of independent component analysis (ICA) on metabolic NBS data, as a multivariate alternative to the traditional methods based on univariate reference ranges established on prior knowledge of the disease biomarker behavior and percentile statistics. The structure of the ICA scores with the aim of investigating their potential to elucidate metabolic relations of the probands is addressed and a comparison with the traditional methods is provided.

We applied ICA on data obtained in a routine newborn screening program performed between 2013 and 2022 in the Czech metabolic screening center in Olomouc. The reason for choosing ICA is that it maximizes non-Gaussianity in the data while searching for independent components, which is beneficial in our context where atypical observations are usually acknowledged regarding the Gaussian distribution. Therefore, we suggest considering the mean ± 5sd rule for the computed independent components as an alternative to the univariate reference ranges of the original variables (i.e., metabolites). Moreover, ICA could potentially reveal other unknown important metabolite ratios theoretically usable in classical newborn screening.

## 2. Materials and Methods

### 2.1. Patients and Samples

The anonymized samples were processed as separate discovery (years 2011–2020, 10,213 disease-free controls and 77 patients suffering from nine different diseases) and validation (years 2021–2022, 150 controls and 20 patients; see Table 1) studies. Blood samples were collected from newborns to the screening cards (Whatman 903) 48–72 h after birth and transported to the laboratory. Samples were prepared according to the CE-IVD kit from Chromsystems (order no. 57000). Discs (3 mm diameter) were punched out from the cards and placed in the 96-well microtiter plate. The extraction buffer (100 µL) with internal standard was added to each sample. The plates were covered with foil, shaken (600 rpm) for 20 min at laboratory temperature, and then centrifuged (10 min at 2000 rpm). The supernatant was used for the analysis.

### 2.2. DBS Analysis by Mass Spectrometry

Mass spectrometric analysis was performed using LC-MS API 4000 Ultimate 3000 RS (Sciex). Amino acids and acylcarnitines were determined in dry blood spots using the above-mentioned CE-IVD kit. The analytical method is routinely used in the laboratory for screening more than 30,000 newborns per year and is accredited according to ISO 15,189 and participates in ERNDIM and Newborn Screening Quality Assurance Program (NSQAP) quality control schemes. Concentrations of 26 amino acids and acylcarnitines used for screening purposes in the Czech Republic were determined and used for further analyses (Appendix A). Data were centered and normalized by log-ratio (clr) transformation without any additional pre-processing steps.

### 2.3. Data Analysis—ICA

The statistical processing of the data is based on ICA and was done with the help of the software R [13] using fICA [14] and robCompositions [15] packages. ICA in general looks for a set of latent variables $z_{i}$, which have the form of linear combinations of the measured variables (metabolites) $x_{k}$. The main property of the latent vector z is that its components are standardized, independent, and likely to follow a non-Gaussian distribution looking as non-Gaussian as possible. Therefore, the method can typically reveal complex sources of outlyingness or groupings within the data [16]. Formally written, the independent component model assumes that the observable *p*-variate vector x is linked with the *p* latent components as Az+b, where A is a nonsingular matrix (i.e., square matrix with non-zero determinant) and b is a location vector, which can be understood as a vector of means of ***x***. The method, in consequence, searches for a matrix W**,** called unmixing matrix, which would identify the latent structures as W(x−b). The resulting vector z is then given uniquely up to the sign and order of its components. There are several strategies on how to estimate the unmixing matrix W; some of them are listed in [17] together with a more detailed description of the method. Our case study is based on the adaptive deflation-based FastICA approach [18] which maximizes the non-Gaussianity of the latent components using the most suitable non-Gaussianity measure for each component separately.

The first step of almost all ICA methods is whitening where the data are standardized and uncorrelated. For that purpose, the covariance matrix of the data needs to be full rank, i.e., nonsingular. Similar to the approach used in [17], we treat the metabolomic data as relative-valued and the final algorithm thus combines ICA with the principles of compositional data analysis [19]. So-called centered log-ratio (clr) coefficients give the natural representation of relative-valued (compositional) data. Within this representation, each of the original variables $x_{k}$ corresponds to one coefficient ${\mathrm{clr}(x}_{k})$ of the form ${\ln(x}_{k}/g(x))$, with g(x) denoting the geometric mean of the whole vector x. This results in a favorable interpretation in the sense of relative dominance of the given part $x_{k}$ over the whole composition (metabolic profile). On the other hand, the construction of the clr coefficients also implies its zero-sum property, which prevents its direct use in the ICA algorithm as it results in a singular data matrix. An alternative representation is given by the family of isometric log-ratio (ilr) coordinates. This representation characterizes the compositional vector x by a system of p−1 orthonormal (i.e., orthogonal, with a unit norm) real-valued coordinates, given as $\mathrm{ilr}\left(x\right)=V^{T}\ln(x)$, with V being a p×(p−1) matrix, called contrast matrix. This coordinate system overcomes the problem of singularity and is, therefore, popularly used in a wide range of multivariate statistical methods, including those relying on the full rank assumption. A detailed description of the construction and properties of this coordinate system is given (e.g., in [19]); however, let us emphasize that the clr and ilr representations are mutually transferable through the contrast matrix V as $\mathrm{ilr}\left(x\right)=V^{T}\mathrm{clr}(x)$. Accordingly, the results obtained from the ilr representation can be transformed into the clr space and interpreted there. Note also here that the clr results are invariant to the chosen ilr basis.

Following the strategy described in [17], first, the unmixing matrix $W^{ilr}$ is estimated for the whitened ilr representation of the data (we used the system of pivot coordinates, see [20] for details) and consequently rotated as $W^{clr}=W^{ilr}V^{T}$. The rows of $W^{clr}$ can be then understood as loading vectors, specifying the contribution of the clr coefficients (i.e., the relative dominance of the respective compositional parts) to the overall values of the latent components called scores. A positive loading determines that the increase in the relative dominance of the respective compositional part results in an increase in the score. The increase in the relative dominance of parts with a negative loading corresponds to the decrease in the score. More specifically, the (p−1)-dimensional vector of scores z is for a patient/control sample with measurement x equal to $W^{clr}(\mathrm{clr}\left(x\right)-b^{\mathrm{clr}})$, with $b^{\mathrm{clr}}$ standing for the mean clr vector.

For formal decision making, the component-wise rule $\mathrm{mean}\left(z_{i}\right)\pm \kappa \cdot \mathrm{sd}\left(z_{i}\right)$*,*
i=1, …,p−1 is employed to decide if an observation is in the direction of the *i*-th latent variable (independent component, ICi) appearing as an outlier. We follow here [21] and use median and median absolute deviation (mad) instead of classical estimates of mean and sd as this avoids the masking effects of outliers. Consequently, the estimates of $\mathrm{sd}\left(z_{i}\right)$ are derived from mads as $1.4826\cdot \mathrm{mad}\left(z_{i}\right)$ [22]. The tuning parameter value κ is used to decide how extreme observations are considered outlying. Based on the training data, this value was set to be 5, as the 5sd rule reflects the low proportion of patients within the sample and, in comparison to the 3 or 4sd rule, yields the best separation of patients from controls.

## 3. Results and Discussion

It is to be expected that ICA, by its nature, projects patients with individual diseases in separate ICs, as each of the diseases affects the metabolic profile differently. In order to gain better insight into which roles the individual metabolites play in the calculations and to be able to relate them to the routine criteria based on reference ranges of biomarkers or their ratios, we propose looking at the structure of an “average score”.

Average score ${\bar{z}}_{i}^{j}$ of the *i*-th IC, computed for the *j*-th group of patients with the individual disease or the group of controls can be written as:z¯ij=∑k=1pW[i,k]clrclr(xk)¯j−bkclr=∑k=1paICijk ,
where i=1, …,p−1 is an index of the IC of the interest and *j* ranges over the groups of patients/controls. $W_{\left[i,k\right]}^{clr}$ stands for the entry of the clr unmixing matrix at the position [i,k]; therefore, it gives the value of loading respective to the *i*-th IC and the *k*-th metabolite, k=1, …,p. Finally, ${\bar{clr(x_{k})}}^{j}$ equals the mean clr value of the *k*-th metabolite computed within the *j*-th group, while $b_{k}^{clr}$ denotes its overall mean (the *k*-th entry of the clr mean vector $b^{\mathrm{clr}}$).

Even though it is not a general feature of ICA, which is within its estimation phase completely unaware of the grouping (i.e., unsupervised), based on the above formula, we can understand the loadings as weights allowing for emphasizing the differences among the individual studied groups within the given IC.

Let us further introduce the concept of “an average member of the score structure” (AMSS). To be able to better describe the sources of (possible) outlyingness of the *j*-th group of patients/controls in the direction of the *i*-th latent variable, a system of AMSS values aICijk is computed as $W_{\left[i,k\right]}^{clr}\left({\bar{clr(x_{k})}}^{j}-b_{k}^{clr}\right)$, k=1, …,p. Each of these values quantifies the contribution of the *k*-th metabolite to the respective mean score ${\bar{z}}_{i}^{j}$.

### 3.1. Interpretation of Components

In Appendix A, there are values of AMSS (including classical sd where applicable; calculated as variance of aICijk) computed for all the studied groups in the first 16 ICs (out of 25), which leads to the best separation of patients/controls. In the following paragraphs, AMSS values for those groups which create separated clusters in the given IC (i.e., a control group vs. one of the patient groups) are sorted and compared. This provides information on which metabolites contribute the most to the separation in terms of the clr coefficients of the measurements. Furthermore, the investigation of AMSS across all the studied groups presents an opportunity for a direct comparison of the effects revealed by ICA to the routinely used approach based on NBS biomarkers.

#### 3.1.1. IC1—MCADD × VLCAD/LCHAD/IVA/GAI

In NBS, patients with MCAD are identified by elevated levels of octanoylcarnitine (C8) and its ratio to acetylcarnitine (C8/C2) as primary markers. Other secondary markers C8/decanoylcarnitine (C10), hexanoylcarnitine (C6), and decenoylcarnitine (C10:1) were suggested; however, these biomarkers are only supportive in the decision-making process [8].

The first latent variable IC1 clearly separates the group of MCAD patients, characterized by its highly negative value (Figure 1A). According to the AMSS values, collected in Appendix A, this separation is mainly given by significantly lower aIC1MCADC8 in comparison to the same value computed for the control group, aIC1controlC8. The second primary applied diagnostic biomarker of MCAD is C8/C2 ratio. In our data, values of AMSS for C2 are similar between the patients and healthy controls, aIC1MCADC2 of −0.028 vs. aIC1controlC2 of 0.000, and the sd is very low for both groups (0.012 and 0.010, respectively). In this ratio, C2 performs as a “reference/anchoring metabolite” (unaffected by the disease and reflecting the general status of the organism) whose level is stable between controls and patients. Other metabolites and their ratios are used as secondary diagnostic markers and serve mainly to confirm the diagnosis, such as C6, C10, C10:1, and C8/C10 ratio. The differences in the AMSS values of these metabolites are not as high as for the C8 primary marker (see Appendix A), but they also affect the separation of the two groups. Contrary to C6 and C8, aIC1MCADC10 alone shifts the MCAD group more into positive values, thus worsening the separation of the two groups, although patients show elevated concentration levels compared to healthy controls.

In the loadings table (included in Appendix A), some metabolites with no obvious pathobiochemical connection to MCAD, e.g., tetradecanoylcarnitine (C14) and tetradecenoylcarnitine (C14:1), are also enhanced with a non-zero loading value. It can be observed that the increased loadings correspond to different AMSS values for patients suffering from Very long-chain acyl-CoA dehydrogenase deficiency (VLCAD), i.e., slightly higher aIC1VLCADC14 and aIC1VLCADC14:1. The result of this is a minor separation of the VLCAD group in IC1 (less than 5sd; similarly with aIC1LCHADC14:1 for Long-chain 3-hydroxyacyl-coenzyme A dehydrogenase deficiency (LCHAD), aIC1IVAC5 for Isovaleric acidemia (IVA), and aIC1GAIC5DC for Glutaric aciduria I (GAI) patients), since loadings perform as “general weights” of the metabolites. Hence, it should be noted that by looking only at loadings, as is typically done, for example, for PCA, one may not be able to reveal the sources for the separation of a specific group of patients. In such a case, it is the table of AMSS values, which are computed for each group separately, that helps with a better understanding of the general picture.

#### 3.1.2. IC2—PKU × MSUD

In IC2, there is a clear separation of PKU patients and controls. The routine diagnostic biomarkers based on the logic of the enzyme defects (Phenylalanine hydroxylase (PAH) in classical PKU and other tetrahydrobiopterin-turnover-related hyperphenylalaninemias (Appendix A)) are Phe, and Phe to Tyr ratio. The concentration levels of Phe are increased since Phe is not converted to Tyr by PAH. Therefore, Tyr levels may be disproportional to increased Phe levels in PKU patients (statistically significant decrease in Tyr observed in PKU patients in our data; Welch Two Sample *t*-test, *p* = 1.098 × 10^−7^), which is reflected in Phe/Tyr ratio. In Appendix A, the biggest difference in AMSS values can be observed between aIC2PKUPhe equal to −9.149 and aIC2controlPhe equal to 0.043. Thus, PKU patients are shifted to negative values in the ICA score plot (Figure 1A). Values of aIC2PKUTyr and aIC2controlTyr are almost similar for the two groups (−0.201 for patients and 0.001 for healthy controls, respectively) pointing to Tyr generally working in NBS as a reference metabolite in the Phe/Tyr ratio.

One healthy control sample from NBS is located in the area exceeding the 5sd threshold near the PKU group. This patient has a Phe concentration of 119.99 μmol L^−1^ and Phe/Tyr ratio of 2.04. Since the cutoff values in NBS routine procedure for Phe and Phe/Tyr ratio are, respectively, 120 μmol L^−1^ and 2, the patient did not exceed both parameters and was therefore flagged as “negative” in the screening despite his values being borderline.

A secondary effect in IC2 is a minor separation of Maple syrup urine disease (MSUD) patients, who are further better clustered in IC4 and described under that section. The separation here is due to increased loadings of clr coefficients of metabolites leucine/isoleucine (Xle) and valine (Val) (biomarkers of MSUD). The values of AMSS in MSUD patients compared to all other groups (Appendix A), namely aIC2MSUDXle and aIC2MSUDVal, corroborate the similar nature of this effect as was the case with aIC1VLCADC14 and aIC1VLCADC14:1.

#### 3.1.3. IC3—LCHAD × GAI

Elevated levels of C16-OH, C18-OH, and C18:1-OH are hallmarks of LCHAD deficiency patients in screening programs. These biomarkers are reflected in the reduced values of aIC3LCHADC16−0H, aIC3LCHADC18−0H, and aIC3LCHADC18:1−0H compared to all other groups (Appendix A), which shift the patient’s group into negative values in the ICA score plot (Figure 1B).

GAI patients are separated in the opposite direction compared to LCHAD patients in IC3. Due to the increased concentration levels of GAI biomarker glutarylcarnitine (C5DC) and its positive aIC3GAIC5DC value (Appendix A), the GAI patient is well separated from all other patients and the control group (Figure 1B) in the positive scores.

#### 3.1.4. IC4—MSUD × IVA

In IC4, the separation of MSUD and IVA patients can be observed. MSUD patients are characterized by increased Xle and Val values. The positive AMSS values of the biomarkers in these patients, aIC4MSUDXle and aIC4MSUDVal (Appendix A), compared to the other groups result in their separation from the mass by shifting them further to the positive values in ICA score plot (Figure 1B).

Among the highest AMSS values, there is also an increased aIC4MSUDC5 which corresponds to a reduction in C5 concentrations in MSUD patients (statistically significant decrease; Welch Two Sample *t*-test, *p* = 2.311 × 10^−5^). With respect to C5 having the highest negative loading value in IC4 (see Appendix A, note the opposite sign of C5 and Xle and Val loadings), it significantly contributes to the resulting high scores of MSUD patients compared to the rest of the data. Moreover, the C5/Xle ratio has been previously described [23] as a ratio detected by the CLIR Productivity Tools [24] in the data from NBS for MSUD in the Netherlands. The authors concluded that “the C5/Xle ratio is predominantly determined by the Xle concentration, and is not of added value”. This statement is slightly in disagreement with our results, where the effect of a reduction in C5 concentration itself can be observed. Thus, the C5/Xle ratio detected by CLIR may be a suitable parameter for the screening.

IVA patients show elevated C5 values. Strongly negative loading of C5 shifts the respective aIC4IVAC5 value as well as the IC4 scores of the IVA patient to the negative values. IVA is described in more detail under section IC8 where the patient is separated without the loadings being highly influenced by other diseases.

#### 3.1.5. IC5—Weights

The clustering visible in IC5 (Figure 1C) was described in a previous publication [17], where loadings of C16, Val, C18:1, C18OH, and C0 discriminated patients with low birth weight. This finding is not at all straightforward and further research is necessary.

#### 3.1.6. IC6—GAI

The highly negative value of aIC6GAIC5DC for the analyte C5DC (primary routine NBS biomarker increased in GAI patients) causes a clear separation of the GAI patient in the ICA score plot from all other groups (Figure 1C, Appendix A). The second screening measure routinely used in NBS to diagnose GAI patients is C5DC/C16 ratio (Table 1). The aIC6jC16 (where *j* ranges over all studied groups) is very similar among all patients and controls, showing C16 together with low sd as one of the viable “reference metabolites” in this component (Appendix A).

In the opposite direction to the GAI patient, one PKU patient exceeds 5sd due to low C5DC concentration.

#### 3.1.7. IC7—ASA

In IC7, a patient with Argininosuccinic aciduria (ASA; detectable in NBS based on increased levels of Citrulline (Cit) and Argininosuccinate (ArgSucc)) is separated. AMSS values of the two associated analytes, aIC7ASACit and aIC7ASAArgSucc, are higher for this patient compared to all other groups, assigning the ASA patient a positive score in ICA score plot (Figure 1D, Appendix A).

#### 3.1.8. IC8—IVA

In NBS, IVA patients are detected using increased levels of C5 concentration and C5/C8 and C5/C2 ratios. The only distinctly separated patient on the *y*-axis of Figure 1D is an IVA case. The negative aIC8IVAC5 value compared to other groups and similar values for “reference metabolites” C8 (except for the group of MCAD patients, where it is a substrate of defective enzymatic reaction; see Section 3.1.1) and C2 across the groups, aIC8jC8 and aIC8jC2, support the use of these analytes in denominators of the biomarker ratios in NBS (Appendix A).

#### 3.1.9. Other ICs

Starting from the independent component 9, score plots (Appendix A) show a spread distribution of healthy control samples with no apparent distinguishing of screened disorders vs. controls (there are no patients exceeding the 5sd threshold). This behavior reflects the fact that components from ICA are post-computationally ordered according to their kurtosis value. Due to this sorting, the first ICs show distributions with heavy tails and are, therefore, expected to find outlying values or unequal sized groups, while the last ICs indicate light-tailed distributions. It seems that the boundary for distinguishing individual groups separated from the mass is located just in IC9 for this data set.

Only a few controls are exceeding 5sd which, however, cannot be explained from the limited data legally available in the NBS program (i.e., birth weight, sampling age, sex, and a restricted set of acylcarnitines and amino acids measured).

### 3.2. Principal Component Analysis

Principal component analysis (PCA), as a gold standard of the dimension reduction methods, was applied on the data (projection using the first two components is given in Figure 2). In order to respect the relative nature of the data, the compositional version of PCA, based on the clr representation, was used [25]. The variance explained by the first components is quite high given the sample size (cumulatively around 40% for the first two PCs; Appendix A; on the other hand, the ability to identify individual groups of patients is rather poor. If the “usual metabolomic approach” was applied, where only the first two or three PCs are usually plotted, a mere two of the nine IEMs present in our data would be identified as separate clusters. Therefore, all pairwise combinations of the total 25 principal components are depicted in a scatter plot (Appendix A).

Groups of LCHAD and MCAD patients are identified in the first two components; IVA is an outlier visible in the PC5, ASA patient is separated in PC10, and GAI can be observed in PC11. On the other hand, the biggest group of PKU patients is not clearly clustered apart from the main group of controls in any of the components. The most visible separation of the PKU group is in the combination of PC13 and PC15 where the patients still partially overlap with controls. This corresponds to the continuity of Phe and Tyr values (i.e., there is no gap between the patient and control groups). MSUD patients are best separated from the main cloud in the combination of PC13 and PC14, while patients suffering homocystinuria due to deficiency of N(5,10)-methylenetetrahydrofolate reductase activity (MTHFR) and VLCAD patients are not clearly separated at all (see also Section 3.3). The overall separation of individual groups is much worse for PCA than for ICA. This is to be expected as non-Gaussianity, the criterion of ICA, is much more natural for outlier detection than variation, the criterion of PCA. Deflation-based ICA as used here can also be seen as a projection pursuit (PP) method and its connection to be a blind estimator of the linear discriminant was recently shown in [26].

### 3.3. Comparison of Diagnostic Performance with the Routine Screening Procedure

False positive rates (FPRs) of ICA using cutoff values based on standard deviations 3sd, 4sd, and 5sd together with FPR of the routine screening procedure are given in Table 2. In our data, an FPR with the tuning parameter set to 4sd provides results comparable to the routine screening procedure [27] while 5sd yields better diagnostic performance (FPR of 0% for all diseases except PKU with FPR of 0.0089%). However, patients and controls in the study are defined based on the metabolic diagnostic criteria used, and this does not guarantee a “correct” classification. Therefore, the two methods cannot be properly compared on the basis of FPR since it is not possible to retrospectively test and attempt to diagnose patients labelled as “false positive” using the ICA method, as these are healthy subjects according to standard screening and it is not ethically acceptable.

Increasing the cutoff value to obtain a better FPR increases the potential risk of false negative results in the data. In the course of a prospective study, it will therefore be necessary to establish “decision cutoff values for individual components” for the projected diseases. The interplay between sensitivity and selectivity and the importance of their proper adjustment is clearly visible in VLCAD and MSUD patients. By raising the threshold to 4sd, one VLCAD patient is not clearly separated (IC1, Figure 1A). Similarly, when shifting to 5sd, one MSUD patient is not correctly identified (IC4, Figure 1B). The special case is the patient with MTHFR, who is not clearly separated in any component constructed by other diseases and does not form a separate component in the discovery phase. As a diagnostic metabolic change, patients exhibit increased levels of homocysteine (hCys) and mildly elevated Met, and Met/Phe ratio. In routine screening, these patients are selected for Met, and Met/Phe ratio with relatively low cutoff. Approximately 1–2% of the patients admitted for the screening are selected for following second-tier testing based on the measurement of hCys level as a specific biomarker compared to methionine. It is expected that the patient with MTHFR could not be detected since hCys is lacking within the screening, and there was only a mild elevation in nonspecific methionine.

ICA could be classified as an interpretable machine learning method that is increasingly used in newborn screening [28]. From that point of view, it could be emphasized that deciphering repetitive biochemical anomalies detected in components, with no apparent association with screened diseases, in prospective study could improve understanding and efficiency of newborn screening.

### 3.4. Validation Study

The validation study included a total of 20 patients suffering from three metabolic disorders diagnosed in our NBS center in 2021 and 2022 as well as 150 healthy control subjects. In the given period, our screening center diagnosed 16 patients with PKU, 3 patients with MCAD, and 1 patient with ASA (Table 1). A validation study was performed using the calculated IC loadings from the discovery data to confirm the ability of the method to diagnose new patients with given IEMs (Appendix A).

A full 5sd separation of these diseases was observed in the respective ICs only (i.e., IC1, IC2, and IC7; Figure 3 and Appendix A). The pattern of the AMSS values (Appendix A) of the most discriminating metabolites for given groups of patients and controls always shows a high degree of similarity with Appendix A. In IC1, MCAD patients were separated from disease-free controls mainly due to decreased aIC1MCADC6 and aIC1MCADC8. Low values of aIC2PKUPhe resulted in a nice separation of PKU patients and control samples in IC2. Increased aIC7ASACit and aIC7ASAArgSucc values for the ASA patient compared to aIC7controlCit and aIC7controlArgSucc indicate his separation in IC7.

The results strongly suggest that the IC loadings together with AMSS values are generally applicable between measurements and thus can be used for the diagnosis of unknown samples in the NBS program.

Given the stability of the results of the ICA method, the metabolite structures identified by AMSS values reflect known interrelationships among biomarkers used in NBS. This allows for a deeper understanding of the pathobiochemistry of the diseases and avoids the necessity of recomputations with each NBS sample batch.

As emphasized in Section 3.3, comparison of diagnostic performance of the method with the traditional approach can only be done by long-term analysis of differentially classified patients.

## 4. Conclusions

The newborn screening program is based on predefined ranges/cutoff values of biomarker molar concentrations. In contrast, ICA is a multivariate statistical method using information about all the metabolites measured. Employing the introduced ICA approach, the coordinate structures calculated from the discovery data were applied to the validation data set containing patients suffering IEMs, together with disease-free control samples. Since the positive patients of the study were selected based on reference methods, it could be considered as a “gold standard” here. ICA detected all the positive patients in the validation study while providing lower FPR (using 5sd threshold). Long-term application of ICA paralleled with the standard methods in the newborn screening setting will provide further insight into the practical utility of the method. However, in general the results demonstrated the potential use of this method as an alternative approach in routine newborn screening.

## Figures and Tables

**Figure 1 IJNS-09-00060-f001:**
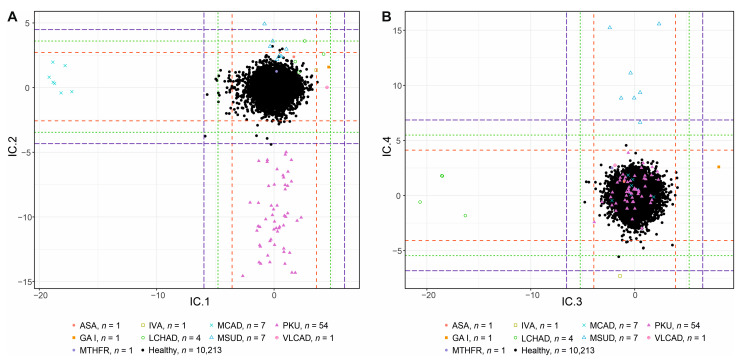
ICA score plots for components IC1 to IC8 of the discovery study: (**A**) Components IC1 and IC2. (**B**) Components IC3 and IC4. (**C**) Components IC5 and IC6. (**D**) Components IC7 and IC8. The dashed lines represent 3 (red),4 (green), and 5sd (blue) rule thresholds.

**Figure 2 IJNS-09-00060-f002:**
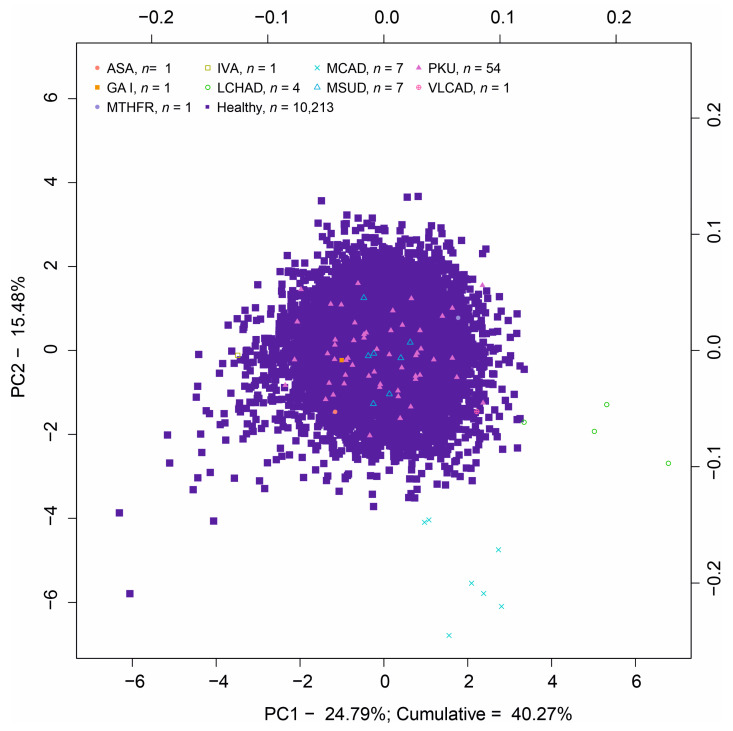
PCA score plots for components PC1 and PC2.

**Figure 3 IJNS-09-00060-f003:**
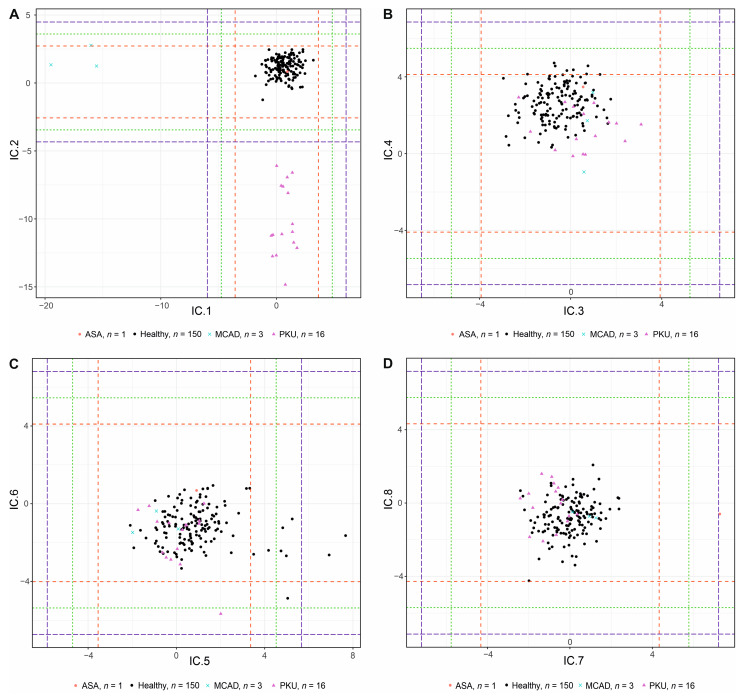
ICA score plots for components IC.1 to IC.8 of the validation study. (**A**) Components IC1 and IC2. (**B**) Components IC3 and IC4. (**C**) Components IC5 and IC6. (**D**) Components IC7 and IC8. The dashed lines represent 3 (red),4 (green), and 5sd (blue) rule thresholds.

**Table 1 IJNS-09-00060-t001:** List of diseases measured using mass spectrometry included in newborn screening in the Czech Republic.

Disease *	Primary Biomarkers	Secondary Biomarkers	Number of Patients (Discovery/Validation Set)
PKU	Phe, Phe/Tyr	-	54/16
MSUD	Xle, Xle/Ala, (Xle + Val)/Pro + Tyr)	Val	7/0
MCAD	C8, C8/C2	C10, C10:1, C6, C8/C10	7/3
LCHAD	C16-OH, C18:1-OH	C18-OH	4/0
VLCAD	C14:1, C14:1/C16	C14	1/0
CPT I	C0, C0/(C16 + C18)	C18, C18:1, C16	-
CPT II/CACT **	C16, (C16 + C18:1)/C2	C18, C18:1, C0	-
GA I	C5DC, C5DC/C16	C5DC/C8	1/0
IVA	C5, C5/C8	C5/C2	1/0
HCY(CBS)	Met, Met/Phe	-	-
HCY(MTHFR)	Met, Met/Phe,	-	1/0
ARG	Arg, Arg/Orn, Arg/Phe	-	-
CIT/ASA **	Cit, Cit/Phe, Orn/Cit, ArgSucc	-	1/1

* For a complete list of OMIM numbers, names of diseases, and their incidence see Appendix A. Full names of metabolites: Phenylalanine (Phe), Tyrosine (Tyr), Leucine/Isoleucine (Xle), Alanine (Ala), Valine (Val), Proline (Pro), Octanoylcarnitine (C8), Acetylcarnitine (C2), Decanoylcarnitine (C10), Decenoylcarnitine (C10:1), Hexanoylcarnitine (C6), 3-Hydroxypalmitoylcarnitine (C16-OH), 3-Hydroxyoleoylcarnitine (C18:1-OH), 3-Hydroxystearoylcarnitine (C18-OH), Tetradecenoylcarnitine (C14:1), Palmitoylcarnitine (C16), Tetradecanoylcarnitine (C14), Carnitine free (C0), Stearoylcarnitine (C18), Oleoylcarnitine (C18:1), Glutarylcarnitine (C5DC), Isovalerylcarnitine/Methylbutyrylcarnitine (C5), Methionine (Met), Arginine (Arg), Ornithine (Orn), Citrulline (Cit), Argininosuccinate (ArgSucc). ** two distinctive metabolic diseases indistinguishable by screening biomarkers.

**Table 2 IJNS-09-00060-t002:** Comparison of false positive rate of ICA method with standard screening evaluation.

IC	Group	−3SD FPR ICA	−3SD FPR ICA	−4SD FPR ICA	−4SD FPR ICA	−5SD FPR ICA	−5SD FPR ICA	FPR David et al., 2019 [27]	FPR Olomouc 2017–2021
IC 1	MCAD	26	0.255%	4	0.039%	0	0.000%	0.002%	0.000%
IC 2	PKU	11	0.108%	4	0.039%	1	0.010%	0.027%	0.003%
IC 3	LCHAD	3	0.029%	0	0.000%	0	0.000%	0.000%	0.001%
IC 4	IVA	7	0.069%	1	0.010%	0	0.000%	0.008%	0.004%
IC 5	weights *	0	0.000%	0	0.000%	0	0.000%	---	---
IC 6	GAI	19	0.186%	2	0.020%	0	0.000%	0.003%	0.003%
IC 7	---	0	0.000%	0	0.000%	0	0.000%	---	---
IC 8	IVA	4	0.039%	0	0.000%	0	0.000%	0.008%	0.004%
**IC**	**Group**	**+3SD FPR ICA**	**+3SD FPR ICA**	**+4SD FPR ICA**	**+4SD FPR ICA**	**+5SD FPR ICA**	**+5SD FPR ICA**	**FPR David et al., 2019** [27]	**FPR Olomouc 2017–2021**
IC 1	VLCAD/IVA/GAI	1	0.010%	0	0.000%	0	0.000%	0.007/0.008/0.003%	0.008/0.004/0.003%
IC 2	MSUD	3	0.029%	0	0.000%	0	0.000%	0.010%	0.019%
IC 3	GAI	3	0.029%	0	0.000%	0	0.000%	0.003%	0.003%
IC 4	MSUD	1	0.010%	0	0.000%	0	0.000%	0.010%	0.019%
IC 5	weights *	123	1.204%	85	0.832%	41	0.401%	---	---
IC 6	---	0	0.000%	0	0.000%	0	0.000%	---	---
IC 7	CIT **	3	0.029%	0	0.000%	0	0.000%	0.006%	0.014%
IC 8	---	2	0.020%	0	0.000%	0	0.000%	---	---

FPR David et al., 2019: [27] The percentage of newborns with a final negative result (after recall—repeated DBS sampling) was presented as the false positive rate (FPR). Data collected from newborn screening laboratories in the Czech Republic (2010–2017). FPR calculated from 888,891 samples. FPR Olomouc 2017–2021: total FPF consisting of FPR after the first measurement and FPR after recall. FPR calculated from 160,951 samples. * In this IC, patients with birth weight below 1500 g are separated [17], ** CIT David et al., 2019: FPR calculated from 181,396 samples (2016–2017).

## Data Availability

The data presented in this study are available on request from the corresponding author. The data are not publicly available due to privacy.

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
