# Peer review of "Multivariate Independent Component Analysis Identifies Patients in Newborn Screening Equally to Adjusted Reference Ranges"

_2409-515X, 2023, doi:10.3390/ijns9040060_

Round 1
Reviewer 1 Report
This contribution confirms that a multivariate analysis of newborn screening markers improves the false positive and false negative rates that can occur by considering only single values of the newborn screening markers. This is a critically important area of investigation.
The authors use the technique of multivariate independent component analysis (ICA) to analyze and compare the results of testing with single marker values in several newborn disorders.
Overall the figures showing the multivariate plots are clear and convincing.
However, since this is a clinical journal, and not a statistical journal, the statistical presentation could be improved. For example, in lines 141 through 145, terms such as "non-singular matrix", "location vector", and "unmixing vector" are introduced but not defined.
Perhaps a table of statistical term could be added, in an appendix if not in the text.
This is an important potential contribution to improvement of newborn screening, and it suggests the value of additional and further studies using multivariate analyses of markers for newborn disorders.
Reviewer 2 Report
This is an interesting study using ICA in NBS to improve diagnostic process and lower the false positive rate. Overall, the use of ICA is interesting.
I have some questions:
1) Which Software was used? According to reference 15 this could be R with some extra packages. But this should be clarified by authors.
2) There is only sparse information about the ML pipeline: Which metabolites were used? Table S2 suggests 25? Are there any pre-processing steps? Normalization etc? The authors should add a paragraph or figure to describe this pipeline, at least as a supplement.
3) Line 15 suggests 15 IEMs, but in Table 1 I count 13?
4) Is there any information why 16 independent components were used?
5) It's impressive that 100% of the validation data was correctly classified. As the author's already suggest, this needs to be further prospectively evaluated. But from a practical point of view, what do the author's suggest in a diagnostic process?
E.g. one must look at the IC1 to diagnose PKU, IC7 for ASA etc?
6) Several studies suggest, that the advantage of ML methods in NBS seems to be the reduction of false positives: false positives may scared parents and increase overload in the NBS lab. But one big challenge in NBS is, that we do not want false negatives, and in traditional cut-off based NBS we try to avoid false negatives by re-analyzing a suspicious probe. Because the are no false positives in the present method (when using 5sd), this may increase the chance of false negatives. This is not the scope of the present paper, but I think this might be an issue in the near future, when ML methods become more present in NBS labs. Maybe the authors have their own thoughts about this and like to share these in the discussion.
Round 2
Reviewer 2 Report
From my point of view, the authors have answered my questions very well, and therefore I recommend the publication of this interesting manuscript.